# Impact of a Teacher Intervention to Encourage Students to Eat School Lunch

**DOI:** 10.3390/ijerph191811553

**Published:** 2022-09-14

**Authors:** Hannah R. Thompson, Stephanie S. Machado, Kristine A. Madsen, Renata Cauchon-Robles, Marisa Neelon, Lorrene Ritchie

**Affiliations:** 1School of Public Health, University of California Berkeley, Berkeley, CA 94720, USA; 2Nutrition Policy Institute, Division of Agriculture and Natural Resources, University of California, Oakland, CA 94607, USA; 3San Francisco Unified School District, San Francisco, CA 94102, USA; 4UC Cooperative Extension, Concord, CA 92250, USA

**Keywords:** school lunch, teacher intervention, school lunch perceptions, school lunch participation, secondary schools

## Abstract

While school meals are often the healthiest option for students, lunch participation remains relatively low. Few approaches for increasing participation have leveraged teachers’ potential social influence. We determined if a teacher intervention about the benefits of school lunch could improve teachers’ perceptions of, and participation in, school lunch, and encouragement of students to eat school lunch. This repeated cross-sectional study included teacher/student survey administration in spring of 2016 and 2018 in 19 public secondary schools (9 intervention, 10 comparison) educating students of ages ≈ 11–18. Intervention teachers received monthly newsletters; lunch taste tests; and a promotional video and website. Mixed effects models with a random effect for school showed the proportion of teachers that reported eating with students increased in intervention schools relative to control schools (difference-in-change: 7.6%; 95% CI: 3.578%, 14.861%), as did student agreement that adults at their schools encouraged them to eat school lunch (difference-in-change: 0.15 on a 5-point scale; 95% CI: 0.061, 0.244). There were no between-group differences in teachers’ perceptions of school meals or teachers’ lunch participation. These findings suggest that teachers’ perceptions of school meals do not necessarily need to improve to promote the school lunch program to students. However, to see meaningful change in teacher lunch participation, the taste of school meals likely needs improving.

## 1. Introduction

A healthy diet during childhood is critical for preventing chronic disease [1], yet the majority of US youth do not meet the Dietary Guidelines for Americans [2]. Schools offer a unique opportunity to intervene to improve youth dietary intake; the National School Lunch Program (NSLP) is in the vast majority of public schools and accessible to students of all socioeconomic backgrounds [3]. Although school meals are often the healthiest option for students [4,5], lunch participation remains relatively low with an average 22.6 million meals served daily across the U.S. [6], despite a pre-K–12 public school enrollment of nearly 49.5 million students [7]. There are many approaches to increasing student participation, but few have focused on leveraging the social influence of in-school adults [8].

Adults [9,10,11,12] are known to influence youth dietary intake through modeling or encouragement. Youth whose parents eat [9,10,12] or encourage [12] a healthy diet are more likely to consume healthy foods. Students have also reported that social support from their friends, family, and teachers [13], as well as role-modeling behaviors of adults, enhances their likelihood of eating healthy foods [11]. It is not clear, however, if these adult influences apply in the school lunch context, and particularly, if teachers can positively influence youth participation in the NSLP. Although youth believe they make healthier dietary choices if adults at school encourage them to do so, it remains unknown if teachers can influence youth dietary intake in school.

Teachers have historically reported neutral or negative perceptions of school lunch programs [14,15], making it unlikely that they would encourage students to participate. However, the Healthy, Hunger-Free Kids Act of 2010 (HHFKA) [16] made school meals healthier [17], and the resulting changes may be a leverage point for improving teacher perceptions. It is important to understand if teacher perceptions can be shifted, and in turn, positively impact teacher modeling behaviors—such as eating school lunch and eating with students in the cafeteria—and increase teachers’ encouragement of student participation in school lunch. Within the context of a larger, multipronged intervention designed to increase student school lunch consumption [18], this study explored the impact of the teacher outreach component on teacher perceptions, modeling, and encouragement related to school lunch. This research tests a conceptual model hypothesizing that outreach to teachers about the school lunch program would improve teacher perceptions of school meals and increase their verbal promotion of the program to students.

## 2. Materials and Methods

This repeated, cross-sectional, quasi-experimental study was part of a larger multipronged school lunch promotion intervention conducted in an urban school district in California over 3 school years (2015–16 through 2017–18) [18]. All 24 traditional middle and high schools in the district participated in the parent study (12 intervention and 12 comparison). Before the study’s onset, five schools piloted intervention components and were thus assigned to the intervention arm. The remaining 19 schools were randomized for intervention (n = 12 total, 5 from pilot testing group and 7 from randomization) or comparison (n = 12), based on the following: school type (middle/high); high vs. low need (based on FRPM eligibility); proportion of white students; and the school’s Academic Performance Index score, a California state measure of academic performance [19]. The intervention, developed from a partnership between the school district, a local design firm, and San Francisco Unified School District, aimed to improve the school dining experience for students and staff. The intervention involved a cafeteria redesign, school meal sales through vending machines and mobile carts, and outreach to teachers to promote the school lunch program [15]. The present study uses anonymous teacher and student survey data from spring 2016 (baseline) and spring 2018 (follow-up). The study received approval by the UC Berkeley Committee for the Protection of Human Subjects (# 2014-12-710) and the school district.

### 2.1. Participants

All teachers from the 24 school sites (N ≈ 700 in intervention and N ≈ 600 in comparison schools) were invited to take part in an anonymous survey in spring 2016, 2017, and 2018. Researchers delivered paper surveys to a point person at each school who was responsible for survey distribution and collection. Teachers were eligible for a gift card raffle for each year of survey administration. Surveys from 2017 were not included in the analyses as the intervention was not fully implemented that school year [15]. A complete case analysis was used; schools with fewer than 4 teacher responses at baseline or follow-up (3 intervention schools (1 middle, 2 high) and 2 comparison high schools) were dropped from this analysis. There were no statistically significant differences in school-level demographic characteristics between schools that were included in this analysis and those that were dropped. The final teacher survey sample included data from 9 intervention and 10 comparison schools (Figure 1).

As part of the larger intervention [15], students were invited to participate in an anonymous survey. A parent notice with an opt-out slip was sent home with each student. School staff administered surveys in homeroom classes. All 7th–10th grade students (N ≈ 13,600) in spring 2016 and 2017 and all 8th–10th grade students (N ≈ 10,600) in spring 2018 were eligible to complete the survey. Only surveys from the baseline (2016) and follow-up (2018) were included in this analysis. One school did not provide student survey data at baseline and was dropped. Surveys from students in schools that were dropped from the teacher survey data sample (n = 5 schools) were also dropped (n = 5103 student surveys). Neither the teacher nor student survey included personal identifiers, precluding the linkage of individual responses between timepoints.

### 2.2. Intervention and Conceptual Framework

The teacher outreach intervention was informed by the Social Learning Theory [20], based on the idea that teachers may promote a change in student behavior through modeling the desired behavior. Prior to this intervention, the district had improved their school meals. Using money raised through local bonds, they changed their meal provider in 2013 to an innovative company providing “kid-inspired chef crafted” meals. Further, they banned all competitive food sales and modified their wellness policy to be more reflective of recommendations from the HHFKA [16]. The teacher outreach intervention was designed to highlight both the school meal changes made prior to the intervention due to the HHFKA and the district’s improvements to the cafeteria environment and school dining experience; to inform school staff about the healthfulness of school meals; and to encourage both adult and student lunch participation.

Teacher outreach included: (1) staff meeting presentations about the school lunch program; (2) video screenings showcasing school meals, innovative serving line models and redesigned cafeterias; (3) taste tests of school lunch menu items; (4) monthly newsletters (n = 9) about topics such as lunch menus, the importance of school meals for student health, achievement, and equity, and ways to encourage student participation in the lunch program; (5) a promoted teacher resource website with information about the school lunch program and classroom activities aligned with academic subjects to promote learning about school meals; (6) and Teacher Appreciation Week promotional materials (e.g., flyers and thank you banners in staff lounges). On average, at intervention schools, taste tests were offered at one staff meeting and at least eight times in the cafeteria or break room during the follow-up study year (2017–2018), reaching an average of 49 teachers/school. Outreach videos were promoted through emails to all intervention teachers and shown at school-wide staff meetings (n > 200 views). If a teacher wanted to eat a school lunch on a typical school day, they had to purchase it themselves at a cost of USD 4.00. A research grant covered teacher outreach funding, as well as 50% of the salary for a research staff member to support the study from within the district. Comparison schools received the same food as intervention schools, but no intervention components.

Figure 2 outlines the pathways believed to change teacher perceptions and behavior, and ultimately, student behavior. Although not specific targets of the intervention, we posited two additional pathways that could result from teachers’ improved perceptions: (1) increased visibility of teachers in the cafeteria eating their lunch with students; and (2) increased teacher school lunch participation. Although this framework hypothesizes that the distal effects of outreach to teachers is increased student participation in school lunch, the intent of this paper is to explore the intervention’s impact on proximal (teacher perceptions of school meal quality) and intermediate (teacher encouragement of student school lunch participation, teacher school lunch participation, and teachers eating lunch with students in cafeteria) outcomes.

### 2.3. Measures

#### 2.3.1. Teacher Survey

Teachers were asked if they agreed, on a 4-point scale from strongly disagree (1) to strongly agree (4) (with an option of “NA/unsure”), that “school meals taste good,” “school meals are healthy,” and “students think school meals are healthy.” Responses were collapsed into either strongly agree/agree or strongly disagree/disagree (“NA/unsure” responses were dropped). Teachers were also asked, “this year, how often did you eat in the cafeteria with students” (which did not specify whether the teacher brought their own lunch or purchased the school lunch), “this year, how often did you encourage your students to eat the school lunch,” and “this year, how often did you eat school lunch?” These questions had five response options (never, ≤1 time per month, 2–3 times per month, 1–3 times per week, and ≥4 times per week) that were coded as never vs. ≥1 time per month (see Appendix A for a copy of the teacher survey).

#### 2.3.2. Student Survey

One question [21] asked: “adults at school encourage me to eat school lunch” with response options on a 5-point scale from strongly disagree (1) to strongly agree (5).

#### 2.3.3. School-Level Data

School-level demographic characteristics were downloaded from the California Department of Education [22].

### 2.4. Analysis

All analyses conducted were intention-to-treat. To determine if the pre-/post-intervention change in teacher perceptions of, and behaviors related to, school meals differed between comparison and intervention groups, logistic mixed effects models with robust standard errors were used. All models included a group-by-year interaction term, adjusted for school-level student free and reduced-price meal eligibility (a proxy for student socioeconomic status) and school type (middle or high), and included a random effect for school to account for teachers clustered within schools.

For the student survey question assessing teacher encouragement to eat the school lunch, a linear mixed effects model with a random effect for school (to account for the clustering of students within schools) and a group-by-year interaction term was used to assess the difference in change between comparison and intervention groups. The outcome was treated as continuous as responses were normally distributed. School-level covariates included free and reduced-price meal eligibility and enrollment and student-level covariates included race/ethnicity, school type (middle and high school), and gender. Grades 7, 9, and 10 at baseline and grades 8 and 10 at follow-up were included in the analysis so as not to violate the assumptions of independence in our statistical models.

## 3. Results

School, student, and teacher characteristics can be found in Table 1. The final analytic sample included 556 (283 intervention, 273 comparison) teacher surveys from 2016 and 533 (226 intervention, 307 comparison) from 2018. No statistically significant differences were seen between intervention and comparison schools in teacher response rates.

A total of 5524 (2503 intervention, 3021 comparison) student surveys from 2016 and 3534 (1514 intervention, 2020 comparison) surveys from 2018 were included in the analysis. No significant differences were seen in student survey response rates between groups; among all students eligible for the survey in the 19 schools which also had teacher survey data, the mean response rate for intervention schools was 65% (baseline) and 62% (follow-up) and for comparison schools it was 62% (baseline) and 54% (follow-up).

No significant school-level differences in demographic characteristics were seen between groups at baseline; however, significant differences in student-level race/ethnicity and gender were seen between student survey respondents in intervention and comparison schools, with intervention schools having a greater proportion of African American students (*p* = 0.002), Latino students (*p* < 0.001), and male students (*p* < 0.001), and comparison schools having a greater proportion of white students (*p* < 0.001), female students (*p* = 0.001), and other gender students (*p* = 0.022).

At baseline, in all schools combined, 86% of teachers reported never eating lunch in the cafeteria with students, 72% reported never eating school lunch, and 43% reported never encouraging students to eat school lunch. Further, at baseline, 68% of teachers agreed/strongly agreed that school meals are healthy, 45% agreed/strongly agreed that school meals taste good, and 22% agreed/strongly agreed that students think school meals taste good. At baseline, 17% of students agreed that adults at school encourage them to eat school lunch.

At follow-up, 14% of teachers in intervention schools reported receiving intervention materials 2.5 times per month or more in the prior year, approximately 58% reported receiving materials 1 time a month or less, and 29% reported never receiving materials.

Table 2 reports the differences in change in frequency of teacher perception and behavior outcomes between intervention and comparison schools from baseline (2016) to follow-up (2018). There were no between-group differences from baseline to follow-up in the proportion of teachers agreeing that school meals taste good (difference-in-change: 7.5%; 95% CI: −5.532%, 20.441%), are healthy (difference-in-change: −2.7%; 95% CI: −16.253%, 10.8095%), or that students think school meals are healthy (difference-in-change: −2.4%; 95% CI: −13.079%, 8.370%).

Although a significant decrease in the proportion of teachers who said they ate with students in the cafeteria was seen in comparison schools, the proportion stayed the same in intervention schools (difference-in-change: 7.6%; 95% CI: 3.578, 14.861%, *p* < 0.01). There were no significant differences in changes in the proportion of teachers reporting eating school lunch or encouraging students to eat school lunch. Compared to their peers at comparison schools, however, students at intervention schools had a relative increase in agreement that adults at their schools encouraged them to eat school lunch (difference-in-change in proportion of students who agree/strongly agree: 4.8%; 95% CI:1.698%, 7.957%, *p* = 0.001; Table 2). In a sensitivity analysis for the student survey question, we included the student survey data from the five schools which had student survey data but did not have teacher survey data; this did not appreciably change the findings (difference-in-change: 0.13; 95% CI: 0.051, 0.211, *p* = 0.002).

## 4. Discussion

This is the first known study to examine both teacher perceptions and behaviors related to school lunch. An intervention designed to promote school lunch consumption appeared to have a modest effect on the teacher-reported frequency of eating in the cafeteria with students and on students’ perceptions of teachers’ encouragement of their participation in the school lunch program in public secondary schools. The teacher outreach intervention, however, did not impact teacher perceptions of school meals or their participation in school lunch.

The intervention was conceptualized, in part, to improve teacher perceptions of school meals, which the district had recently enhanced. It was hypothesized that improved perceptions related to the health and appeal of school meals would lead teachers to eat the lunch themselves and encourage students to eat it as well. However, we did not see changes in teacher perceptions, the first component of our conceptual model (Figure 2). The majority of teachers (70%) across schools/timepoints perceived school lunches to be healthy. Following the HHFKA changes to nutrition standards which began in 2012–2013, school meals have typically become the most nutritious option for students [5]; the relatively high proportion of teachers at baseline rating school meals as healthy may reflect a recognition of these changes. Teacher perceptions of taste, however, were low in all schools at baseline, with only 45% of teachers agreeing that school lunch tastes good, and remained low at follow-up (with 48% agreeing). Anecdotally, in one of the author’s (SM) interviews of the implementation staff in the present study, teachers often mentioned that their negative views about school meals stem from previous experiences with the meal program. Despite recent improvements to school meal quality, the meals remained in the “reheat and serve” style, which relies heavily on pre-packaged and processed foods that are made at a central location and then shipped to schools to be reheated, which have been the source of complaints in prior school lunch studies [23]. Despite outreach which included school lunch taste tests, teachers’ unfavorable taste perceptions of school lunch did not change. A lack of change may indicate that the taste of the meals themselves, a component not addressed by the HHFKA nor by this intervention, needs to be improved to see enhanced teacher perceptions.

Though teachers’ school lunch perceptions did not change, students in intervention schools reported an increase in teachers encouraging them to eat school lunch. The staff meeting presentations and the newsletter components of the intervention included information about the value of the school lunch program to student achievement, health, and equity. Teachers may have increased their encouragement to students, realizing that school meals may be the best meal option available. Anecdotal reports from intervention staff showed that teachers were most engaged when staff presented information about the “big picture” of school meals: connecting school meals with student achievement and student health. Though teachers’ perceptions of the healthfulness and taste of meals may not have changed, they may have seen the value for students in other ways.

We did see significant increases in one of the two teacher modeling behaviors described in the conceptual model; teachers in intervention schools reported a relative increase in the frequency of eating with students in the cafeteria, though we did not assess if teachers were eating their own lunch or school-bought lunch. This finding could also explain the increase in students reporting that teachers encouraged them to eat school lunch. If, as posited above, the intervention enhances teachers’ perceived value of school lunch (such as leading to improvement in student achievement or student health, which we did not test), teachers may increase their visibility in the cafeteria to encourage student participation. Further research is needed to explore the potential viability of the conceptual link between teacher modeling and student meal participation.

Given that teacher perceptions of school meals did not change, it is not surprising that their self-reported frequency of eating school lunch also did not increase. The parent study measuring objective changes in teacher and staff participation found a slight, yet significant, relative decrease in teacher lunch participation in intervention schools (3.0% at baseline and 1.7% at follow-up) relative to comparison schools (1.9% at baseline and 1.1% at follow-up) [24]. A potential driver of teacher participation is the taste of the meals themselves. Although teachers’ perceptions of the taste of school meals did not change, it is possible that teachers’ expectations for the meals were raised by the intervention, and tasting the school meals did not meet their higher expectations. Teachers who agreed that school meals tasted good had a significantly higher frequency of self-reported school lunch participation than those who disagreed. This relationship between perceptions of taste or quality and school meal participation is consistent with other studies looking at student populations [25,26,27]. Findings from our study suggest that an outreach intervention alone is not sufficient for improving teacher school lunch participation; the taste of the meals may need to be addressed as well.

Based on our findings, we have updated our conceptual model (Figure 3), with additions highlighted with dashed lines. The revised model suggests that teacher encouragement and eating in the cafeteria with students does not rely on teachers’ perceptions of school meals’ taste or healthfulness (though these behaviors may further improve if perceptions also improve). Instead, the perceived value of the school lunch program, such as the program’s value in addressing student health or achievement, may mediate the relationships between teacher outreach and teacher encouragement and eating in the cafeteria. This model also suggests that improvements to school meal quality would be necessary to improve teacher perceptions of taste and healthfulness (thus leading to an increase in their participation in the school meal program). Future studies should test this updated model while specifically measuring whether teachers eat the school lunch or their own lunch.

Importantly, with over 70% of teachers reporting exposure to the intervention at least monthly in the follow-up study year (2017–2018), the intervention reach was moderately high for a nutrition intervention in the school setting [28,29]. In an average intervention school, teachers received nine promotional newsletters, saw a short (11 min) promotional video at a staff meeting, and had the opportunity to sample school meals at least nine times. As research funding covered a half-time staff member to implement teacher outreach activities, we do not anticipate that the intervention dose delivered would be higher in other real-world settings.

This study has multiple limitations. The lack of teacher survey participation in five schools limited our ability to make inferences about all study schools. The lack of teacher-level demographic information may have resulted in unmeasured confounding in our teacher data models. Further, the lack of teacher survey identifiers precluded our ability to determine if the samples at baseline and follow-up were independent, thus potentially violating the statistical assumption of independence. The multicomponent nature of the intervention also renders it difficult to identify which intervention components had the greatest impact on teacher behavior. Finally, this intervention explicitly focused on teachers and excluded other school staff (e.g., administrators, food service staff) who could also influence eating behaviors. Future work should be more inclusive of all adults interacting with students in schools.

## 5. Conclusions

This study provides insight into an understudied but important topic: teacher perceptions and behaviors surrounding school lunch. Few studies have explored teacher perceptions of school meals [14,15], with no known intervention studies on the topic. This study contributes to the literature by proposing a conceptual model for testing in subsequent research. As this intervention did not improve teacher perceptions of school lunch’s healthfulness or taste, we were unable to establish a link between such perceptions, the encouragement of students to eat school lunch, and role modeling. However, the results indicate that teacher perceptions of school meals do not necessarily need to be improved for teachers to promote the program to students. In addition, findings suggest that an outreach intervention alone is not sufficient for increasing teacher school lunch participation; the taste of school meals likely also needs to be improved. Future studies should explore determinants of teacher perceptions and values related to school meals as potential levers for improving school lunch participation.

## Figures and Tables

**Figure 1 ijerph-19-11553-f001:**
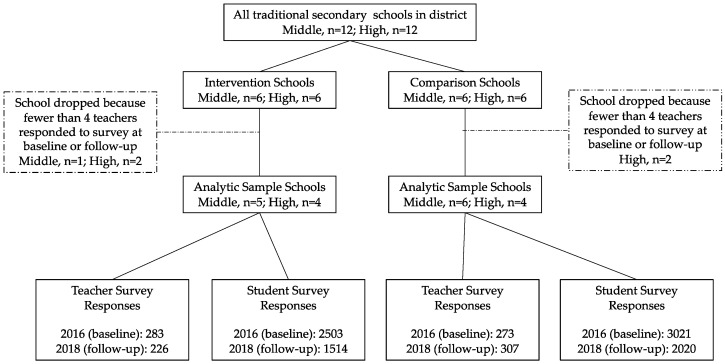
Study flow chart.

**Figure 2 ijerph-19-11553-f002:**
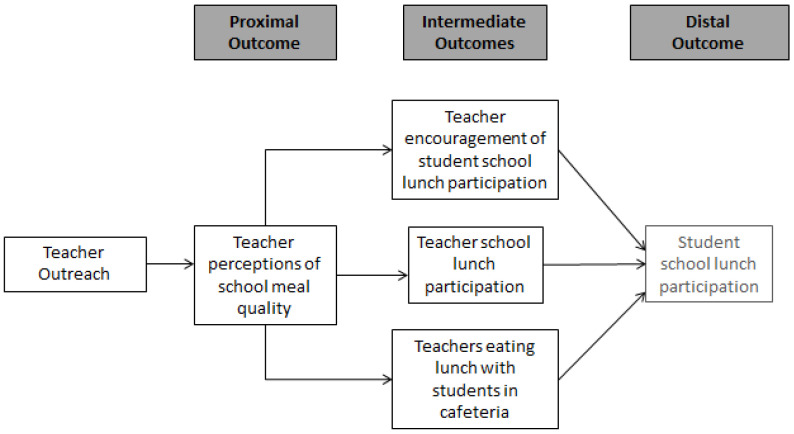
Teacher Outreach Intervention Conceptual Framework. Teacher outreach included newsletters, videos, school lunch tastings, a website, and other promotional materials related to school lunch.

**Figure 3 ijerph-19-11553-f003:**
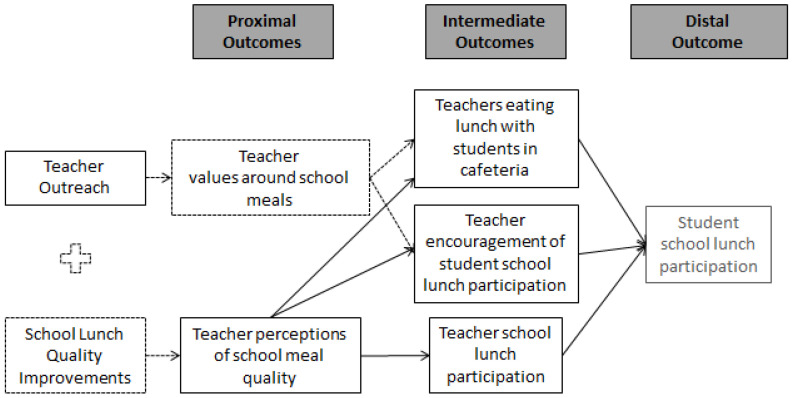
Revised Teacher Outreach Conceptual Model. Teacher outreach included newsletters, videos, school lunch tastings, a website, and other promotional materials related to school lunch. Outcomes in dashed boxes were unmeasured in this study.

**Table 1 ijerph-19-11553-t001:** Baseline (spring 2016) sample school, student, and teacher characteristics.

	Intervention Schools	Comparison Schools	*p*-Value ^a^
School-Level Characteristics	(n = 9 schools)	(n = 10 schools)	
Students enrolled, mean ± SD	929 ± 165	975 ± 212	0.8704
Students eligible for FRPM, % ± SD	68.3 ± 3.2	60.2 ± 4.8	0.1856
Student-Level Characteristics	(n = 2429 students)	(n = 2923 students)	
Baseline response rate per school, % ± SD	64.6 ± 13.4	62.1 ± 16.7	0.7216
Follow-up response rate per school, % ± SD	62.0 ± 23.7	53.7 ± 23.4	0.4524
Student-reported race/ethnicity, % ± SD			
African American	4.5 ± 20.8	2.9 ± 16.8	0.002
Asian	45.8 ± 49.8	46.8 ± 50.0	0.441
Latino	24.0 ± 42.7	16.3 ± 36.9	<0.001
White	4.2 ± 19.9	10.9 ± 31.2	<0.001
Other ^b^	21.5 ± 41.1	23.1 ± 42.1	0.182
Student-reported gender, % ± SD			
Female	42.6 ± 49.5	47.0 ± 49.9	0.001
Male	48.3 ± 50.0	41.9 ± 49.4	<0.001
Other^c^	9.2 ± 28.9	11.1 ± 31.4	0.022
Teacher characteristics	(n = 283 teachers)	(n = 273 teachers)	
Number of teachers employed per school ^d^, mean ± SD	53 ± 24.2	55 ± 32.4	0.882
Baseline response rate per school, % ± SD	58.0 ± 32.8	53.2 ± 28.2	0.7364
Follow-up response rate per school, % ± SD	50.1 ± 25.5	51.2 ± 21.4	0.921

^a^*p*-values from unpaired *t*-tests. ^b^ Other race comprised of other, multiple, or declined to state race/ethnicity. ^c^ Other gender comprised of other and declined to state gender. ^d^ Number of teachers employed in follow-up year (spring 2018); baseline (spring 2016) data were unavailable. IQR: interquartile range.

**Table 2 ijerph-19-11553-t002:** Adjusted between-group difference in changes in teacher-reported perceptions of, and behaviors related to, school lunch ^a^.

	Intervention ^b^			Comparison ^b^			Between-Group Difference in Change ^c^ (95% CI)
	Baseline (n = 283) Mean ± SE	Follow-Up (n = 226) Mean ± SE	Difference Mean (95% CI)	Baseline (n = 273) Mean ± SE	Follow-Up (n = 307) Mean ± SE	Difference Mean (95% CI)	
Teacher perceptions, % who agree/strongly agree							
School meals taste good	46.1 ± 3.5	52.8 ± 3.6	6.8 ± 2.3 (2.321, 11.180)	44.3 ± 5.2	43.6 ± 4.4	−1.0 ± 6.1 (−12.711, 11.303)	7.5 ± 6.6 (−5.532, 20.441)
School meals are healthy	72.5 ± 3.6	74.4 ± 3.0	1.8 ± 4.0 (−6.072, 9.736)	64.1 ± 4.3	68.7 ± 3.5	4.6 ± 5.6 (−6.365, 15.473)	−2.7 ± 6.9 (−16.253, 10.8095)
Students think school meals are healthy	22.6 ± 4.1	17.9 ± 2.6	−4.7 ± 4.6 (−13.635, 4.314)	21.0 ± 3.9	18.7 ± 2.9	−2.3 ± 3.1 (−8.360, 3.749)	−2.4 ± 5.5 (−13.079, 8.370)
Teacher behaviors, % who did this ≥1 time per month							
Eat with students in the cafeteria	11.3 ± 2.0	10.7 ± 2.1	−1.0 ± 2.6 (−5.733, 4.425)	15.7 ± 3.0	7.4 ± 1.4	−8.3 ± 2.3 (−12.869, −3.658)	7.6 ± 3.7 (3.578, 14.861)
Encourage students to eat school meals	57.0 ± 6.4	57.6 ± 5.1	1.0 ± 7.0 (−13.032, 14.228)	60.8 ± 3.3	60.3 ± 4.0	−1.0 ± 2.8 (−5.919, 4.889)	1.1 ± 7.5 (−13.592, 15.818)
Eat school lunch	27.1 ± 3.1	29.2 ± 2.5	2.2 ± 2.6 (−2.849, 7.359)	29.1 ± 4.0	23.3 ± 2.1	−5.8 ± 3.5 (−12.625, 1.102)	8.0 ± 4.4 (−5.176, 16.550)
Student perceptions, % who agree/strongly agree							
Adults at school encourage me to eat school lunch	15.7 ± 1.4	16.9 ±1.5	1.3 ± 1.2 (−1.079, 3.645)	17.5 ± 1.3	14.0 ± 1.4	−3.5 ± 1.1 (−5.604, −1.485)	4.8 ± 1.6 (1.698, 7.957)

^a^ Values for teacher perception questions from logistic mixed effects models with robust standard errors. Values for teacher behaviors from generalized linear models with a Gamma family log link. All models accounted for clustering by school and adjusted for school-level free and reduced-price meal eligibility and school type (middle/high). ^b^ Sample sizes vary slightly by question and timepoint. Missingness range for both intervention and comparison schools <1–3%. Some teachers may have taken the survey at both baseline and follow-up. ^c^ Change from baseline to follow-up in intervention schools compared to comparison schools.

## Data Availability

Data associated with this article are available by request to the authors. Please contact Hannah R. Thompson at ThompsonH@Berkeley.edu.

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
