# Peer review of "Impact of a Teacher Intervention to Encourage Students to Eat School Lunch"

_ijerph, 2022, doi:10.3390/ijerph191811553_

Round 1
Reviewer 1 Report
Impact of a Teacher Intervention to Encourage Students to Eat School Lunch
This is a well-written manuscript that aimed to examine the impact of a teacher intervention package on teachers’ perceptions of, and participation in, school lunch, and encouragement of students to eat school lunch. Although the intervention had no significant impact on teacher perceptions of school meals or teachers’ lunch participations, the authors found a modest effect on teacher-reported frequency of eating in the cafeteria with students. There was also a positive impact on student’s perception of teachers’ encouragement of their participation in school lunch program. Considering the importance of school meals in meeting nutrient needs of children, especially those with limited access to healthy nutritious meals, strategies to increase consumption of these meals are important and encouraged. Please find below a few comments and suggestions to the authors:
1. Were there any differences in characteristics between schools that were included in the analysis and those that were not included?
2. Lines 79-83 mention that 5 schools were dropped from the analysis. However, lines 90-92 mention that student surveys from “schools that were dropped” were also not included in the analysis and lists the “n” as 4 schools. This does not match with the initial “n=5” stated. Can the authors double check their numbers to ensure they tally and are consistent?
3. No information is given about how the intervention and comparison groups were selected. Was it random? Were they selected purposefully? Any information would be helpful.
4. A flow diagram showing the numbers from recruitment to analysis would be helpful and make things much clearer. I’d recommend that authors include a flow chart.
5. Lines 199-203: Authors should be consistent in stating the “n” for each outcome reported. They can choose to either report the “n” or not but should be consistent.
6. Was the analysis an “intention-to-treat” analysis?
7. Lines 204-207: The authors report that 29% of teachers in intervention schools reported never receiving intervention materials. Do the authors think this impact the results in anyway? The authors could run a sensitivity analysis to try to examine if it made a difference.
8. Lines 213: The sentence “…students think school meals taste good…”. It should be “students think school meals are healthy…” and not “taste good” as per table 2. Authors should double check to ensure they are reporting correctly.
9. Lines 228-234: Can the authors include the student survey results as part of Table 2? Especially since it is one of the main findings they report.
10. Lines 231-233: Authors state that a sensitivity analysis did not change the “findings”. Authors should be more specific which of the findings they are referring to.
11. 260-264: Great point!
12. Lines 294 – 296: Is this finding from the study dataset? There was no mention of that in the results section.
13. Figure 2: Authors should please include a key for figure 2 to show what the dashed and solid lines stand for.
Author Response
- Were there any differences in characteristics between schools that were included in the analysis and those that were not included?
We looked at potential school-level demographic differences for school enrollment, proportion of students who qualify for free or reduced-price meals, proportion of African American students, proportion of Asian students, proportion of Latino students, and proportion of white students and found no statistically significant differences. Therefore, we have added the following sentence to section 2.1 on Participants: “There were no statistically significant differences in school-level demographic characteristics between schools that were included in this analysis and those that were dropped.”
- Lines 79-83 mention that 5 schools were dropped from the analysis. However, lines 90-92 mention that student surveys from “schools that were dropped” were also not included in the analysis and lists the “n” as 4 schools. This does not match with the initial “n=5” stated. Can the authors double check their numbers to ensure they tally and are consistent?
Thank you for catching this mistake and our apologies. The correct number of schools dropped is 5 and we have corrected this.
- No information is given about how the intervention and comparison groups were selected. Was it random? Were they selected purposefully? Any information would be helpful.
We have added information about how schools were selected to be in either the intervention or comparison groups: “Before study onset, five schools piloted intervention components and were thus assigned to the intervention arm. The remaining 19 schools were randomized to intervention (n = 12 total, 5 from pilot testing group and 7 from randomization) or comparison (n = 12), based on the following: school type (middle/high); high vs. low need (based on FRPM eligibility); proportion of white students; and the school’s Academic Performance Index score, a California state measure of academic performance.18”
- A flow diagram showing the numbers from recruitment to analysis would be helpful and make things much clearer. I’d recommend that authors include a flow chart.
Thank you for this suggestion. We have added a flow diagram showing the numbers from recruitment to analysis (new Figure 1).
- Lines 199-203: Authors should be consistent in stating the “n” for each outcome reported. They can choose to either report the “n” or not but should be consistent.
Thank you for pointing our this inconsistency. We have removed the n’s so that we are reporting uniformly throughout the methods section.
- Was the analysis an “intention-to-treat” analysis?
Yes – all analyses were intention-to-treat. We have now clarified that in section 2.4 Analysis.
- Lines 204-207: The authors report that 29% of teachers in intervention schools reported never receiving intervention materials. Do the authors think this impact the results in anyway? The authors could run a sensitivity analysis to try to examine if it made a difference.
This is a great thought – however because we ran an intention-to-treat analysis, which is the strongest analytic approach given this was a quasi-experimental study (with most schools randomized to treatment/control), we would like to present this intention-to-treat data only.
- Lines 213: The sentence “…students think school meals taste good…”. It should be “students think school meals are healthy…” and not “taste good” as per table 2. Authors should double check to ensure they are reporting correctly.
Our sincerest thanks for catching this typo. We have corrected it to read “students think school meals are healthy”, which is reported correctly.
- Lines 228-234: Can the authors include the student survey results as part of Table 2? Especially since it is one of the main findings they report.
Absolutely. We have added this to Table 2. For ease of interpretation of the data added to the table, we have moved from reporting mean score for the student survey question (scale 1-5) to reporting the proportion of students who agree/strongly agree) so the interpretation is easier and also aligned with how we present the teacher survey perception questions.
- Lines 231-233: Authors state that a sensitivity analysis did not change the “findings”. Authors should be more specific which of the findings they are referring to.
We have clarified, “In a sensitivity analysis for the student survey question,”
- 260-264: Great point!
Thank you!
- Lines 294 – 296: Is this finding from the study dataset? There was no mention of that in the results section.
We presented in Table 2 that teachers reported a relative 7.2% increase in frequency of eating with students in the cafeteria. However, we did not ask teachers if they were eating their own lunch (brought from home) or if it was a lunch they purchased from the cafeteria.
- Figure 2: Authors should please include a key for figure 2 to show what the dashed and solid lines stand for.
We have added to the figure caption that the outcomes outlined in dashed boxes for for outcomes that were unmeasured in this study.
Reviewer 2 Report
Thank you for the opportunity to review this paper examining the impact of a teacher intervention to encourage students to eat school lunch. To enhance the readability and wider context for the audience, I have some minor comments.
Abstract:
It is worth specifying the age of the students, as it has an impact on the ability of the teacher to influence their behavior. The age of students at secondary school in different countries may vary.
The last two sentences are not clear, it is worth redrafting them. Also the last part of the sentence "he taste and appeal of school meals likely need improving" does not result from the study.
Introduction:
line 35: Please provide some numbers, e.g. percentage of students eating regularly lunch at school if you state that “lunch participation remains relatively low”
Materials and methods:
line 67: I don't understand what it means in this context: “BLINDED FOR REVIEW”
lines 74-75: Does the number apply to teachers or students? It is not clear.
General question: Are lunches for teachers fully paid or (partially) subsidized?
Discussion/conclusions:
Do the authors believe that there is a need for teachers to eat lunch with students more often at school? The survey results indicate that it is not necessary for teachers to encourage students to eat lunch at school. So maybe there is no such need?
General: It is worth including a questionnaire for teachers in the additional materials.
Author Response
Abstract:
It is worth specifying the age of the students, as it has an impact on the ability of the teacher to influence their behavior. The age of students at secondary school in different countries may vary.
Thank you for this point. We have added to the abstract that secondary schools educate students ages»11-18.
The last two sentences are not clear, it is worth redrafting them. Also the last part of the sentence "he taste and appeal of school meals likely need improving" does not result from the study.
Apologies for that – we were trying to stay in the abstract word count, which contributed to the lack of clarity. We have modified the second to last sentence to read, “These findings suggest that teachers’ perceptions of school meals do not necessarily need to improve to promote the school lunch program to students.” We have also modified the last sentence to read, “However, to see meaningful change in teacher lunch participation, the taste of school meals likely needs improving.” which is directly related to our findings.
Introduction:
line 35: Please provide some numbers, e.g. percentage of students eating regularly lunch at school if you state that “lunch participation remains relatively low”
This is something we really struggled with. Unfortunately, there are not good estimates of the proportion of students who participate, nationally, on a daily basis. Part of what makes the calculation so challenging is that daily attendance differs from enrollment, so there’s no stable denominator for the calculation. To try and give a better sense of what the proportion could be, without doing a calculation that would be incorrect, we have added the following, which hopefully helps address your concern: “While school meals are often the healthiest option for students,4,5 lunch participation remains relatively low, with an average 22.6 million meals served daily across the U.S,6 despite pre-K-12 public school enrollment of nearly 49.5 million students.”
Materials and methods:
line 67: I don't understand what it means in this context: “BLINDED FOR REVIEW”
Our apologies for that. In many journals they do not want you to say which institution reviewed the IRB protocol, because it could give away who the authors are. Therefore, we redacted that information and replaced it with “BLINDED FOR REVIEW” so that the reviewers are no aware of which institution we work at.
lines 74-75: Does the number apply to teachers or students? It is not clear.
General question: Are lunches for teachers fully paid or (partially) subsidized?
Great question. Our apologies for not including this information. We have added the following to our methods section: “If a teacher wanted to eat a school lunch on a typical school day, they had to purchase it themself at a cost of $4.00 USD.”
Discussion/conclusions:
Do the authors believe that there is a need for teachers to eat lunch with students more often at school? The survey results indicate that it is not necessary for teachers to encourage students to eat lunch at school. So maybe there is no such need?
This is a great question. We still believe that eating lunch with students in the cafeteria to be one of the intermediate outcomes on the potential pathways between teacher’s perceptions of school meal quality and student lunch participation, however it is certainly not the only intermediate outcome. You are right that our results suggest that in certain contexts, this may in fact not be a necessary intermediate outcome!
General: It is worth including a questionnaire for teachers in the additional materials.
This is a great suggestion. We have added our teacher survey to the supplementary materials.
Reviewer 3 Report
You have done a relevant and interesting study. Some changes would increased the quality of your paper:
Introduction: Please give more background information about the students (health risks, eating habits and suggestibility by age group, gender, ses, etc.)
Methods: The student survey is quite smal (just one question). Are there more data available (as the study is part of a larger intervention)? Also more information about the service would be interesting (time for eating, choices between several menus, quality of food etc.)
Results/ Conclusion: the student-reported race/Ethnicity differ significantly at baseline. Please discus possible impact on the results.
Author Response
Introduction: Please give more background information about the students (health risks, eating habits and suggestibility by age group, gender, ses, etc.)
Thanks so much for this suggestion. We wish we were able to do this. We unfortunately do not have more background information on these students, other than the sociodemographic data we have already provided in the paper.
Methods: The student survey is quite smal (just one question). Are there more data available (as the study is part of a larger intervention)? Also more information about the service would be interesting (time for eating, choices between several menus, quality of food etc.)
Yes – there is more data from the student survey. You make a great point. This was the only question we asked on the student survey that was directly related to the teacher intervention. We have published another paper on the findings from the student survey. That paper also includes more information on food service!
Results/ Conclusion: the student-reported race/Ethnicity differ significantly at baseline. Please discus possible impact on the results.
While there were statistically significant differences in the school-level proportion of students who were Latino (24% vs 16%) and White (4% vs. 11%) at baseline, we do not think these are appreciably different enough to impact the findings from the one student survey question we included in this analysis (Adults at school encourage me to eat school lunch). However, in case there was any potential confounding from this, we also included student race/ethnicity in the statistical model where we analyzed the student survey question, so hopefully this would help address any confounding.